# APPA Increases Lifespan and Stress Resistance via Lipid Metabolism and Insulin/IGF-1 Signal Pathway in *Caenorhabditis elegans*

**DOI:** 10.3390/ijms241813682

**Published:** 2023-09-05

**Authors:** Shiyao Wang, Dongfa Lin, Jiaofei Cao, Liping Wang

**Affiliations:** Key Laboratory for Molecular Enzymology and Engineering of Ministry of Education, School of Life Sciences, Jilin University, Changchun 130012, China; shiyao21@mails.jlu.edu.cn (S.W.); lindf21@mails.jlu.edu.cn (D.L.); caojf1321@mails.jlu.edu.cn (J.C.)

**Keywords:** APPA, lifespan, stress resistant, insulin/IGF-1, *Caenorhabditis elegans*

## Abstract

Animal studies have proven that 1-acetyl-5-phenyl-1H-pyrrol-3-yl acetate (APPA) is a powerful antioxidant as a novel aldose reductase inhibitor independently synthesized by our laboratory; however, there is no current information on APPA’s anti-aging mechanism. Therefore, this study examined the impact and mechanism of APPA’s anti-aging and anti-oxidation capacity using the *Caenorhabditis elegans* model. The results demonstrated that APPA increases *C. elegans*’ longevity without affecting the typical metabolism of *Escherichia coli* OP50 (OP50). APPA also had a non-toxic effect on *C. elegans*, increased locomotor ability, decreased the levels of reactive oxygen species, lipofuscin, and fat, and increased anti-stress capacity. QRT-PCR analysis further revealed that APPA upregulated the expression of antioxidant genes, including *sod-3*, *gst-4*, and *hsp-16.2*, and the critical downstream transcription factors, *daf-16*, *skn-1*, and *hsf-1* of the insulin/insulin-like growth factor (IGF) receptor, *daf-2*. In addition, *fat-6* and *nhr-80* were upregulated. However, the APPA’s life-prolonging effects were absent on the *daf-2*, *daf-16*, *skn-1*, and *hsf-1* mutants implying that the APPA’s life-prolonging mechanism depends on the insulin/IGF-1 signaling system. The transcriptome sequencing also revealed that the mitochondrial route was also strongly associated with the APPA life extension, consistent with *mev-1* and *isp-1* mutant life assays. These findings aid in the investigation of APPA’s longevity extension mechanism.

## 1. Introduction

Global population aging is on the rise, with the percentage of persons over 65 gradually rising. At the same time, age positively correlates with serious diseases like cancer, diabetes, and neurological disorders [1]. Therefore, aging is both a social and medical issue [2]. Aging is a natural, irreversible process influenced by genes and environmental factors [3]. The reactive oxygen species (ROS) generated in the body alters the cell shape and vitality, impairing metabolism and immunological function. Mainstream theory holds that telomere, mitochondrial, and DNA damage are directly related to aging [4]. Based on this, developing drugs or treatments that slow the aging process and boost the body’s antioxidant capacity is crucial. *Caenorhabditis elegans* is a popular biological model for research on the mechanisms of oxidative stress and aging, due to its ease to culture, short life cycle of 15 to 21 days, strong ability to reproduce, small size, no ethical limitations, numerous mutant strains, low cost, and high productivity [5,6,7].

Research has revealed that 12 signal pathways and 60–80% of the genes in *C. elegans* are identical to those in humans, including Nrf2/SKN-1 and insulin/insulin-like growth factor (IGF)-1 conserved signal (IIS) pathways involved in aging and oxidative stress [8,9,10,11,12]. Therefore, *C. elegans* growth and development, and its nervous system responses can forecast processes relevant in higher animals, making *C. elegans* a significant player in signal transduction [13,14,15].

Currently, many monomer compounds isolated from natural sources have beneficial anti-aging properties and therapeutic value, such as epigallocatechin gallate which has high antioxidant activity [16]. In addition, previous research has demonstrated the potent antioxidant and free radical-scavenging effects of APPA, a benzida lysine analog, on cells and animals (Figure 1A). Therefore, it is crucial to research how monomer compounds act as anti-aging agents. The present study explored the anti-aging properties of APPA on nematodes. APPA was created and produced in the lab. It has been discovered that APPA, a novel aldose-reductase inhibitor, exhibits strong antioxidant action. Therefore, we investigated the anti-aging and anti-oxidative stress ability and mechanism of APPA on *C. elegans* using it as a model.

## 2. Results

### 2.1. APPA Extended the Lifespan of C. elegans

The *C. elegans* life expectancy was significantly increased by APPA by 3.49, 4.92, and 2.75% following treatment with 0.09, 0.18, and 0.36 mg/mL of APPA (Appendix A), respectively (Figure 1B). Further details for all survival assays are presented in Appendix A. Compared to the control group, APPA had no discernible effect on *E. coli* OP50 development (Figure 1C). Since 0.18 mg/mL APPA maximally extended *C. elegans* lifetime without impairing the growth cycle of *E. coli* OP50, it was chosen for subsequent studies. The impact of APPA on *C. elegans*’ basic physiological characteristics is depicted in Appendix A.

### 2.2. APPA Reduces Lipofuscin Accumulation in C. elegans

Lipofuscin, a naturally occurring fluorescent molecule and a biomarker of aging, was built up with *C. elegans* aging, slowing down the cell function and metabolism. Its fluorescence was green under UV light. Overall, *C. elegans* accumulated less lipofuscin under APPA treatment than the control group (Figure 2A,B).

### 2.3. APPA Reduces Fat Accumulation in Nematodes

According to the oil red O staining assay, fat accumulation in *C. elegans* tissues was drastically decreased under APPA treatment compared to the control (Figure 3A,B), consistent with the findings beforehand, where APPA did not influence *C. elegans* growth and development but improved its motor function (Appendix A). Fatty acid desaturase *fat-6*, as a key target of *nhr-80*, is involved in the metabolism and synthesis of fatty acids [17]. QRT-PCR revealed that APPA elevated *fat-6* and *nhr-80* mRNA levels in *C. elegans* compared to the control group (Figure 3C), implying that *fat-6/nhr-80* significantly contributed to the APPA-mediated suppression of fat storage in *C. elegans*.

### 2.4. APPA Promotes the Expression and Activity of Antioxidant Defense Enzymes, Which Increase Survival and Reduces ROS Levels in Stress-Induced C. elegans

We further confirmed the impact of APPA on *C. elegans*’ SOD and CAT activities. SOD and CAT, endogenous ROS defense enzymes involved in the detoxification process of chemicals generated by oxidation in cells, are made more active by APPA (Figure 4A,B).

An imbalance in ROS generation and clearance is known as oxidative stress, characterized by the death of *C. elegans*, brought on by exogenous oxidants (paraquat) or environmental conditions [18]. Exposure to 50 mM paraquat oxidative stress and 35 °C hot conditions revealed that 0.18 mg/mL of APPA considerably improved *C. elegans*’ capacity to withstand stress (Figure 4D,E).

Paraquat can speed up *C. elegans*’ aging by causing intracellular ROS buildup and high ROS levels. We measured the ROS content in nematodes using the fluorescent probe H_2_DCF-DA in order to further investigate the connection between APPA delaying senescence and ROS increase in *C. elegans*. The outcomes demonstrated that APPA decreased the ROS levels in *C. elegans* treated with paraquat, suggesting that APPA increased nematode lifespan by decreasing *C. elegans*’ ROS levels (Figure 4C,F).

Thus, APPA promotes the expression and activity of antioxidant defense enzymes, which increase survival and reduce ROS levels in stress-induced *C. elegans*.

### 2.5. APPA Does Not Extend the Lifespan of C. elegans through Dietary Restriction

Nutrient availability in the environment affects the mechanisms that control aging in *C. elegans*. Dietary restriction influences signal transduction systems, including IIS and TOR signal transduction as well as anabolic activities like mitochondrial metabolism and AMPK activation [19]. 

A dietary restriction genetic model, *eat-2* (ad1116), was used to investigate if APPA treatment while decreasing the amount of food in *C. elegans* induced malnutrition, lengthened the lifespan of *C. elegans,* and boosted its anti-stress ability [20]. mutant *C. elegans*’s acetylcholine-gated cationic selective channels’ activity was altered, which reduced food intake, limited calories, and ultimately increased lifetime [21].

APPA treatment significantly increased *C. elegans* lifespan compared to the control group (Figure 5A; Appendix A; Appendix A), and did not affect the ability of nematodes to swallow (Appendix A). This implies that the mechanism of APPA’s effect on *C. elegans* lifespan was unrelated to the dietary restriction.

### 2.6. APPA Extended C. elegans Lifespan through the Insulin Signal Pathway

The lifespan assay revealed that *C. elegans* lifespan was increased by mutations in the insulin/IGF receptor, *daf-2*, which was linked to the nuclear translocation of downstream transcription factors *daf-16*, *hsf-1*, and *skn-1*. The findings also demonstrated that APPA did not increase the lifespan of *C. elegans daf-2 (e1370)* and *daf-16 (mgDf50)* (Figure 5B,C). These findings imply that the IIS signal pathway controlled by *daf-2/daf-16* was essential for the APPA life extension mechanism. Moreover, qRT-PCR revealed that APPA increased the *daf-16* mRNA level in *C. elegans* compared to the control group, suggesting that *daf-2/daf-16* was crucial for the life extension of *C. elegans* (Figure 5F).

### 2.7. SKN-1 Was Required for APPA-Mediated Lifespan Extension

QRT-PCR revealed that APPA upregulated the *skn-1* gene’s expression as an antioxidant regulator (Figure 5F). However, APPA did not lengthen the life of *skn-1 (zu67) C. elegans* (Figure 5D). The two assays revealed that the SKN-1/Nrf2 pathway was essential for the APPA life extension mechanism.

### 2.8. HSF-1 Was Required for APPA-Mediated Lifespan Extension

APPA increased the expression of the *hsf-1* gene but did not increase the lifespan of the *C. elegans hsf-1 (sy441)* strain (Figure 5E,F). These findings imply that *hsf-1* was crucial in the life-extension mechanism of APPA.

### 2.9. APPA Enhances the Expression of Stress Resistance Gene in C. elegans

QRT-PCR revealed that APPA increased superoxide dismutase-3 (*sod-3*), glutathione S-transferase-4 (*gst-4*), and heat shock proteins-16.2 (*hsp-16.2*) mRNA expression levels in *C. elegans* compared to the control group (Figure 6C). *Sod-3* protein expression was considerably upregulated according to the quantitative fluorescence analysis, implying that APPA boosts the expression of genes that shield *C. elegans* from oxidative and thermal stress (Figure 6A,B). Overall, increased stress resistance mediated by APPA and the extension of *C. elegans* longevity depended on the overexpression of stress resistance genes.

### 2.10. The Longevity Extension of APPA Depends on DAF-16/FOXO

An evaluation of the *C. elegans daf-16 (mgDf50)* mutant lifespan to establish whether the APPA-induced longevity of *C. elegans* was reliant on *daf-16* revealed that 0.18 mg/mL of APPA did not increase the lifespan of the *daf-16* mutant (Figure 5C). The findings demonstrated that for an increase in *C. elegans*’ lifespan, APPA was dependent on the DAF-16/FOXO pathway. An assessment of the nuclear localization of *daf-16* before and after administration using TJ356 transgenic strain *C. elegans* harboring the DAF-16::GFP gene revealed that *daf-16* trafficking into the nucleus was considerably activated by 0.18 mg/mL of APPA (Figure 7A,B). In addition, the expression of *daf-16* and its downstream antioxidant target genes were required for APPA to boost anti-stress capabilities and postpone *C. elegans* aging. *Daf-16* deficiency also prevented APPA from successfully extending longevity since APPA was dependent on *daf-16*.

### 2.11. Effect of APPA on the Whole Genome Transcription Profile of C. elegans

Using the *C. elegans* gene database as a reference, transcriptome sequencing analysis revealed 3721 differential genes (FDR < 0.01 and Fold Change ≥ 2), of which 1548 were significantly upregulated and 2173 significantly downregulated (Figure 8A). The top 10 genes upregulated by treatment with 0.18 mg/mL of APPA are shown in Appendix A.

GO analysis revealed that the upregulated genes were enriched in three categories: molecular function, cellular component, and biological process, primarily in cellular processes, cell biology, and catalytic activity (Figure 8D). 

These findings suggested that growth regulation and energy intake may be involved in the lifespan mechanism mediated by APPA. Additionally, APPA treatment influenced the expression of genes related to oxidative phosphorylation, implying that APPA depends on mitochondrial pathways to increase *C. elegans* longevity.

Furthermore, KEGG analysis revealed that the genes were primarily enriched in the *C. elegans* longevity-regulating pathway, endocytosis, MAPK signal pathway, ribosome, axon regeneration, calcium signal pathway, mTOR signal pathway, Wnt signal pathway, FOXO signal pathway, and fatty acid metabolism (Figure 8E). The lifespan regulatory pathway highly elevated differentially enriched genes, including SOD-3, GST-4, sHSPs, LIPs-17, FAT-6, ClpP, and TIM23 (Figure 9).

### 2.12. Lifespan Extension by APPA Depends on the Mitochondrial Pathway

Whole transcriptome sequencing revealed that mitochondria-related pathways were dramatically altered in the APPA-treated group compared to the control group. APPA treatment also increased the expression of anti-stress genes, preventing ROS build-up in the mitochondria. These findings suggested that the mechanism by which APPA increased the life cycle of *C. elegans* depended on the mitochondrial pathway since it did not enhance the lifespan of *isp-1 (qm150)* and *mev-1 (kn1) C. elegans* (Figure 8B,C).

## 3. Discussion

### 3.1. Effects of APPA on Longevity and Physiological Indices of C. elegans

APPA has a good antioxidant effect. *E. coli* OP50 inactivation and the timing of its development and metabolism influence the lifespan of *C. elegans* [22]. In addition, *C. elegans* longevity depended on the *E. coli* OP50 growth schedule. However, it did not affect the growth rate and metabolism of *E. coli* OP50 at 0.18 mg/mL, which had effective anti-aging effects. *C. elegans* fertility and lifespan are occasionally inversely associated [22]. Nonetheless, 0.18 mg/mL of APPA did not affect *C. elegans*’ reproduction ability. Generally, the neurological system, muscular system, and mitochondrial function positively influence *C. elegans* longevity. Moreover, *C. elegans* physiology is impacted by the pertinent regulation of various systems and processes [23,24]. When measuring physiological indices, APPA can considerably enhance *C. elegans* mobility without negatively affecting its size, body width, or food intake (Appendix A). Herein, APPA administration was performed on the fifth day of the adult worm since that is when the improvement impact was the strongest.

As *C. elegans* ages, alterations in the body’s lipid metabolism may lead to fat build-up and the ROS are increased with lipid accumulation [21]. In the present study, the fat content in nematodes was decreased by APPA treatment. The *nhr-80* and *fat-6* upstream molecules significantly contribute to the lipid metabolism pathway. *Fat-6* plays a key role in innate immunity. On the other hand, *nhr-80* is a member of the NHR transcription factor family in *C. elegans*, including RNA polymerase II, involved in the positive control of transcription and fatty acid metabolism [25]. The amount of lipofuscin that tends to build up in *C. elegans*’ intestinal cells indicates how old the organism is [26]. In this study, the lipofuscin levels in *C. elegans* were dramatically lowered following APPA therapy.

### 3.2. APPA Enhances the Stress Capacity of Nematodes

Regular cell metabolism generates ROS [23]. Excessive ROS levels induce physiological harm, significantly impacting physiology, altering gene expression, interfering with mitochondrial function, and impairing cell signal transduction [23,27,28]. Collectively, this accelerates aging. The ROS also generates other free radicals, including superoxide and hydrogen peroxide, as a by-product of the oxidative phosphorylation by the mitochondria [29,30].These free radicals easily interact with the DNA, proteins, and lipids due to their high reactivity, leading to oxidative damage [27,31]. The accumulation of oxidative damage is one of the primary causes of aging, according to the free radical aging theory. APPA reduced mitochondria ROS accumulation by activating CAT and SOD activities and enhanced the anti-stress ability of nematodes.

GO and KEGG analyses revealed that the p38 MAPK pathway is present in the *C. elegans* gut. APPA-upregulated genes are enriched in pathways linked to growth control and energy metabolism. *Skn-1* controls the phase II detoxification genes to protect against oxidative stress through the p38 MAPK pathway [32,33,34,35,36,37]. In addition, the mitochondrial pathway may be connected to a mechanism extending life since APPA dramatically increased the mRNA expression of *clpp-1*. More free radicals are generated, and the mitochondrial membrane potential and respiratory capability decrease as *C. elegans* ages [38]. In addition to producing adenosine triphosphate and other metabolic products through the oxidative phosphorylation of glucose and fatty acids, mitochondria also serve as a semi-autonomous organelle and a source of energy for the cell [39,40]. These processes control the cell homeostasis and apoptosis [41]. Growing evidence points to a direct connection between mitochondria and longevity [42,43]. Moreover, the mutant mitochondrial electron transport chain longevity tests revealed that the mitochondrial route is necessary for the APPA life extension.

### 3.3. APPA Extended C. elegans Lifespan through the Insulin Signal Pathway

The IIS has a significant impact on extending the lifespan of *C. elegans* [44,45,46]. Herein, APPA enhanced *C. elegans*’ motor function and the expression of genes linked to the IIS pathway. The insulin/IGF-1 signaling pathway controls *C. elegans*’ reproductive and metabolic capacities as a growth factor signal mechanism [19], while *daf-2* interacts with the insulin-like peptide ligand as a key regulator [20]. *Daf-16*, a FOXO transcription factor that functions downstream of *daf-2* is controlled by phosphatidylinositol 3-kinase and the IIS receptors [47]. *Daf-16* modifies the expression of genes related to stress and metabolic pathways when transferred from the cytoplasm to the nucleus in response to biological stimuli.

Additionally, the activation of *daf-16* and *daf-2* increased mitochondrial muscle function and motor activity. Longevity assays on the matching gene mutations further demonstrated the dependence of the APPA’s life-extension mechanism on the IIS pathway. *Skn-1* controls the oxidative stress response in the p38 MAPK pathway and longevity in parallel with DAF-16/FOXO in the IIS pathway is controlled by *daf-2* [48].In other words, the SKN1/NRF2 pathway is linked to life extension [49,50]. 

An imbalance in ROS generation and clearance is known as oxidative stress, characterized by the death of *C. elegans*, brought on by exogenous oxidants or environmental conditions [18]. Since *C. elegans* lifespan extension is typically accompanied by enhanced anti-stress ability, qRT-PCR detection of anti-oxidative stress and detoxemia genes was performed. APPA dramatically enhanced the mRNA expression of *sod-3*, *gst-4*, and *hsp-16.2*. *Sod-3* stimulates protein isomerization, activity, and scavenging of the superoxide free radicals in the mitochondrial respiration bodies. *Sod-3* and *gst-4* are controlled by crucial transcription factors, *daf-16* and *skn-1*, in IIS [51]. On the other hand, *gst-4* increases glutathione transferase activity, takes part in the manufacture and metabolism of glutathione, reduces ROS, and protects against oxidative stress by detoxifying oxidative stress by-products [52]. Therefore, the *C. elegans* antioxidant defense system is very adaptable in response to stressors. Among these are the tiny HSPs, a class of low-molecular-weight polypeptides largely conserved from yeast to humans [53,54,55,56]. The increase in the tiny *hsp-16.2* reduces the glutathione levels, enhances unfolded protein binding activity, and improves heat stress resistance [57,58]. *Hsp-16.2* is a heat shock protein controlled by *hsf-1*. Proteins are damaged by heat stress, producing toxic oligomers that cause cellular malfunction. The *hsf-1* transcription factor protects against heat shock by controlling the expression of chaperone heat shock proteins and preserving the protein homeostasis under heat stress [59]. Glutathione transferase controls the signal transduction pathway, the breakdown of aromatic amino acids, and detoxification of oxidative stress by-products [60]. Thus, the *hsp-16.2* expression is regarded as a trustworthy sign of heat stress resistance. As a stress-reducer, *hsp-16.2* stimulates glutathione transferase in *C. elegans* [61]. The insulin/IGF-1 signal pathway is necessary for the extended survival of overexpressed heat shock proteins. More importantly, APPA preserves protein homeostasis [62]. *Daf-16*’s downstream transcription factor, *sod-3*, is easily triggered to control antioxidant detoxification. Herein, APPA greatly boosted *sod-3* expression and *daf-16* penetration degree in fluorescence detection of *sod-3* and *daf-16* mutants, consistent with the qRT-PCR analysis. 

## 4. Materials and Methods

### 4.1. Strains

The Caenorhabditis Genetics Center (CGC) provided the N2-Bristol (wild type), *eat-2 (ad1116)*, *daf-2 (e1370)*, *daf-16 (mgDf50)*, *skn-1 (zu67)*, *hsf-1 (sy441)*, CF1553, TJ356, and *Escherichia coli* OP50 (OP50) strains of *C. elegans* (Appendix A). The strains were cultivated at either 20 or 16 °C. At the same time, 0.2% DMSO reserve solution (*v*/*v*) was prepared by dissolving the APPA powder in water and DMSO. Subsequently, APPA and OP50 1:1 at varying concentrations were diluted and evenly coated on nematode growth medium (NGM) boards prior to delivery. Before delivery, OP50 was also doubled and incubated for 13 h at 180 rpm and 37 °C.

### 4.2. Lifespan Assay

The L4 nematodes were synchronized and moved to NGM, containing FUDR (160 μM), OP50 (500 μL), and APPA (0.09, 0.18 or 0.36 mg/mL). Each APPA concentration group had three plates with 100 nematodes each. The plates were incubated at 20 °C for incubation, and every 24 h, the nematodes were transferred to a new NGM with fresh OP50 until all the nematodes had died. The number of alive, dead, and lost nematodes was counted regularly. Dead nematodes were identified by their immobility when softly touched on the head. The lifespan assay was replicated thrice. Appendix A displays the *C. elegans* strains’ life spans.

### 4.3. OP50 Growth Assay

A total of 150 μL of a mixed solution consisting of the LB liquid medium, OP50, and various APPA concentrations (0.09, 0.18, or 0.36 mg/mL) were added in each well of a 96-well plate. Next, the absorbance was measured at 600 nm from 0 to 12 h after every two hours in an incubator maintained at 37 °C using the Infinity 200 Pro Microplate Reader (Tecan, Switzerland). Finally, a growth curve plot was generated using the Image J 15.2v program.

### 4.4. Lipofuscin Assay

On day 12, following treatment with or without 0.18 mg/mL APPA, the nematodes were narcotized using 10 mM levamisole (Aladdin, Shanghai, China). Subsequently, 30 nematodes per group were measured and detected by purple fluorescence with an excitation wavelength of 400–430 nm under the Olympus X71 fluorescent microscope (Tokyo, Japan), viewed through a ten-fold objective. The nematode size was determined from the bright field image. In addition, the relative fluorescence intensity of the nematodes in vivo was measured using the Image J 15.2v program, then the fluorescence intensity of nematodes per unit area was calculated. The data were analyzed in terms of the fluorescence intensity of the experimental group relative to the blank group. 

### 4.5. Body Bend Assay

At the top of the agar, 30 nematodes were put in 10 μL of M9 buffer after 3, 5, or 7 days of treatment with or without 0.18 mg/mL APPA. Next, the number of positive selection curves produced every 30 s was noted.

### 4.6. Pharyngeal Pumping Assay

The pharyngeal pump rate was monitored and recorded using a COIC stereoscope (BK1201, Chongqing, China) every ten seconds in twenty nematodes treated with or without 0.18 mg/mL of APPA for 3, 5, or 7 days.

### 4.7. Fertility Assay

Five synchronized L4 nematodes were cultured on NGM petri dishes supplemented with 0.18 or 0 mg/mL of APPA every 24 h until the end of the oviposition phase. Finally, the eggs were incubated for 48 h at 20 °C in NGM, and the offspring hatched were counted.

### 4.8. Body Length and Body Width Assay

The egg-laying adults were transferred to NGM supplemented with 0.18 or 0 mg/mL of APPA in petri dishes. Subsequently, the adults were separated from the eggs three hours later. A control group consisting of N2 was established. The egg growth at 0, 24, 48, and 96 h was recorded, using ImageJ 15.2v, and their body dimensions were measured.

### 4.9. Oil Red O Staining

The oil red O stain solution was prepared by combining 2% Triton X-100 with 1% oil Red O solution (Aladdin, Shanghai, China). Next, 30 nematodes were treated with 0.18 or 0 mg/mL of APPA for five days, then rinsed in two changes of M9 buffer, fixed for 20 min with the oil red O stain solution, washed again with M9 buffer, and placed on a 2% agarose gel mat. Subsequently, 30 nematodes per group were measured and detected under the Olympus X71 fluorescent microscope (Tokyo, Japan), viewed through a ten-fold objective. In addition, the mean oil red O intensity in nematodes was measured using Image J 15.2v [63].

### 4.10. Quantification of Reactive Oxygen Species (ROS)

The nematodes were rinsed with M9 buffer solution to remove OP50 after a 5-day treatment with 0.18 or 0 mg/mL of APPA. Afterward, 30 nematodes were treated with 100 µM of 2, 7-dichlorofluorescein diacetate (H_2_DCF-DA) (Meilunbio, Shanghai, China), incubated at 35 °C for 30 min, anesthetized with levamisole, and then placed on 2% agarose [52]. Subsequently, 30 nematodes per group were measured and detected by blue fluorescence (Ex 430–455 nm) under the Olympus X71 fluorescent microscope (Tokyo, Japan), viewed through a ten-fold objective. The nematode size was determined from the bright field image. Finally, their relative fluorescence intensity was determined by the integrated density and quantitative fluorescence intensity by ImageJ 15.2v [63]. Then, the fluorescence intensity of nematodes per unit area was calculated. The data were analyzed in terms of the fluorescence intensity of the experimental group relative to the blank group. 

### 4.11. Heat Stress Assay

Thirty nematodes were pre-treated with 0.18 mg/mL APPA and then maintained at 35 °C for 4 h and 20 °C for 12 h. The nematode survival rate was then calculated following the recovery phase [64].

### 4.12. Oxidative Stress Assay

Thirty nematodes were transplanted to NGM plates with 50 mM paraquat (Aladdin, Shanghai, China) after APPA pretreatment [65]. Every 24 h, the survival rate was noted until all worms perished. The criterion for death was that the nematodes were immobile when touched.

### 4.13. Fluorescence Measurement

Thirty nematodes of mutant strain CF1553 were sedated with 10 mM leimidazole (Aladdin, Shanghai, China) and placed on a 2% agar-agar mat after being treated with 0.18 or 0 mg/mL of APPA for five days. Subsequently, the nematode images were captured using the fluorescence microscope (Olympus X71, Tokyo, Japan), viewed through a ten-fold objective. The fluorescence intensity was measured using ImageJ 15.2v [66].

### 4.14. DAF-16 Nuclear Localization Assays

Thirty mutant TJ356 nematodes were sedated with 10 mM leimidazole (Aladdin, Shanghai, China) and placed on a 2% agar-agar mat after being treated with 0.18 or 0 mg/mL of APPA for five days. Subsequently, representative images of transgenic TJ356 nematodes’ green fluorescent protein (GFP) localization in cytoplasm and nucleus were captured by fluorescence microscope (Olympus X71, Tokyo, Japan) by blue fluorescence (Ex 430–455 nm), viewed through a ten-fold objective. In addition, the number of nematodes in each group was counted.

### 4.15. Enzymatic Activity Assays

Nematodes from the control group and the APPA-treated group for five days were homogenized and centrifuged at 5000 g for five minutes. Then, the enzyme activity of the supernatant was determined by the commercial SOD and CAT assay kits (Beyotime Biotechnology, Shanghai, China). The total protein was determined using the Bradford assay, and the enzyme activity was normalized to the protein content.

### 4.16. Quantitative Real-Time PCR

The nematodes received 0.18 or 0 mg/mL of APPA treatment for five days. Next, 2000 nematodes per treatment were used to extract total RNA using the Transzol UP RNA Kit (TransGen Biotech, Beijing, China). Following that, reverse transcribing RNA into cDNA, using the qRT-PCR kit (TransGen Biotech, Beijing, China) according to the manufacturer’s instructions. The cDNA was amplified using a CFX96 Real-Time System (BIO-RAD, Hercules, CA, USA) using TransStart Top Green qPCR SuperMix (TransGen Biotech, Beijing, China). The reaction volume for qRT-PCR was 20 µL, consisting of 10 µM primer and 2 µL cDNA. The predegeneration process was at 95 °C for a period of 5 mins and the denaturation process which required 45 cycles at 95 °C for 5 s, and 60 °C for 30 s for annealing. Ultimately, the reaction was discontinued with the condition of 65 °C for 30 s. The 2^−∆∆CT^ method was used to quantify relative fold changes in transcript levels, with *act-1* expression used as an internal reference to standardize total mRNA levels. Appendix A lists the primers that were utilized.

### 4.17. RNA Sequencing

The gene expressions in wild-type nematodes and wild-type nematodes treated with 0.18 mg/mL of APPA at day five were assessed by the Biomarker Technology Company (Qingdao, China) *n* = 3 (2000 individuals per group), using the WBcel235 genome as a reference. The gene expression was quantified as fragments per kb of the transcript per million fragments mapped as follows:(1)$$FPKM=\frac{cDNAFragments}{MappedFragments\left(Millions\right)\;*\;TranscriptLength(kb)}$$

Next, the differential expression analysis of the two genomes was performed using edgeR, with FDR < 0.05 and fold change ≥ 2 as the thresholds for significant differential expression.

The size of the library insertion fragments is 200–400 bp; using illumina double-ended sequencing, that is, paired end sequencing; the read length is the sequencing mode of PE150, and the read depth is the base number of the sequencing of the project, which is sequenced in accordance with 6G.

### 4.18. Statistical Analysis

All experiments were conducted in triplicate. All data were expressed as mean ± SEM. The GraphPad 7 software was used in the lifespan assay to calculate the *p*-value by log-rank test. The Student *t*-test was used to assess quantitative data at * *p* < 0.05, ** *p* < 0.01, *** *p* < 0.001, and **** *p* < 0.0001 significance levels.

## 5. Conclusions

APPA activates numerous cellular defense pathways and is a promising anti-aging therapy deserving of further studies and development. APPA increases *C. elegans*’ stress resistance and extends its life span; thus, it is a potent antioxidant to enhance human health.

## Figures and Tables

**Figure 1 ijms-24-13682-f001:**
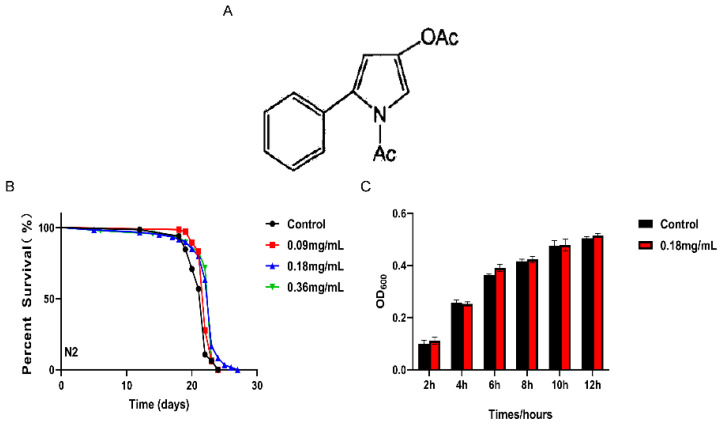
Effects of APPA concentrations on nematode lifespan and growth of *E. coli* OP50. (**A**) The chemical structure of APPA. (**B**) Survival curve of nematodes fed 0, 0.09, 0.18, and 0.36 mg/mL APPA under normal culture conditions of 20 °C, among which APPA 0.18 mg/mL extended the average life span of nematodes by 4.92% compared with the control group. *n* = 3 (100 individuals per group), Kaplan–Meier survival analysis with Log-Rank test. (**C**) APPA did not affect the growth of OP50, which was detected by OD600. Data were analyzed by a Student’s *t*-test using GraphPad 7. Values are presented as the mean ± SEM.

**Figure 2 ijms-24-13682-f002:**
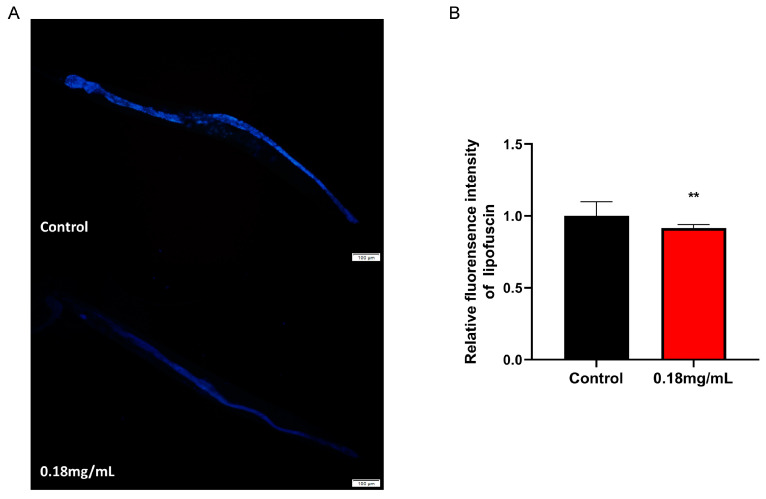
Effects of APPA on lipofuscin accumulation in nematodes. (**A**) Blue autofluorescent lipofuscin fluorescence images in nematodes. *n* = 3 (30 individuals per group). (**B**) Fluorescence quantitation of blue autofluorescent lipofuscin in nematodes. Lipofuscin accumulation in nematodes was measured on the eighth day of administration, and APPA significantly decreased lipofuscin accumulation compared with control. The data were analyzed using a Student’s *t*-test. The values were shown as the mean ± SEM, ** *p* < 0.01.

**Figure 3 ijms-24-13682-f003:**
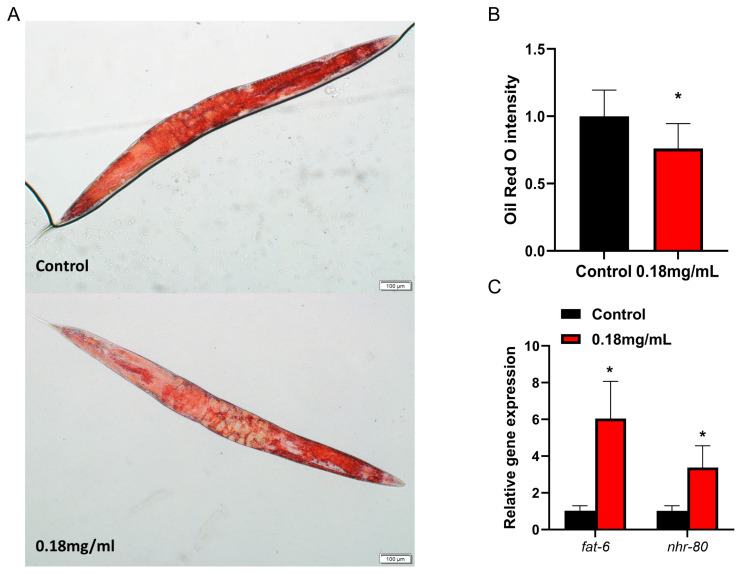
Effects of APPA on fat accumulation and related genes in nematodes. (**A**) Compared with controls. Fat accumulation in nematodes was measured on the fifth day of administration. Oil red O staining diagram under bright field conditions. *n* = 3 (30 individuals per group). (**B**) Quantitative results of nematodes stained with oil red O. Compared with the control group, APPA significantly reduced fat levels in nematodes. (**C**) APPA enhances the expression of *fat-6* and *nhr-80*. Numerical data were analyzed by Student’s *t*-test using GraphPad 7 and values were presented as mean ± SEM, *n* = 3 (2000 individuals per group) * *p* < 0.05.

**Figure 4 ijms-24-13682-f004:**
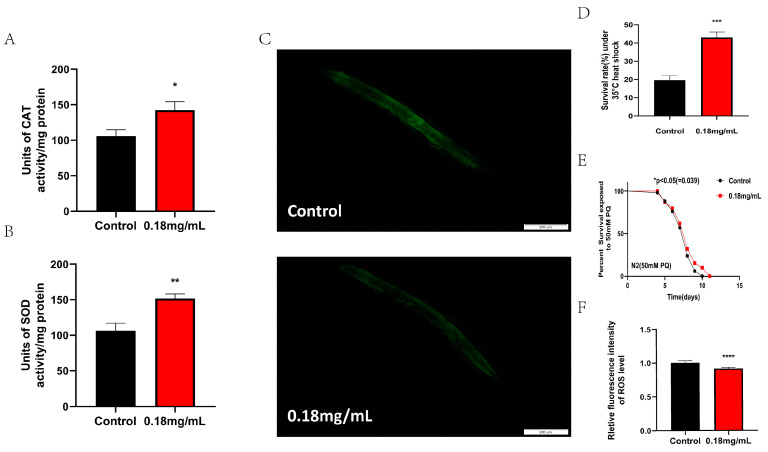
APPA enhanced the antioxidant stress and heat stress resistance of nematodes. APPA enhanced the activity of SOD and CAT. APPA enhanced SOD (**A**) and CAT (**B**) activity in nematodes treated for five days. Data in bar graphs are expressed as mean ± SEM. (* *p* < 0.05, ** *p* < 0.01, two-tailed Student’s *t*-test). (**C**) Fluorescent images of ROS levels in nematodes. *n* = 3 (30 individuals per group). (**D**) Survival rate of nematodes under 35 °C heat stress for 5 h. *n* = 3 (30 individuals per group).*** *p* < 0.001 (**E**) Survival rate of nematodes under oxidative stress of 50 mM PQ. Numerical data were analyzed by Log-rank test and values were presented as mean ± SEM, * *p* < 0.05. *n* = 3 (30 individuals per group). (**F**) Fluorescence quantification of ROS in nematodes. The data were analyzed using a Log-rank *t*-test. The values were shown as the mean ± SEM, **** *p* < 0.0001.

**Figure 5 ijms-24-13682-f005:**
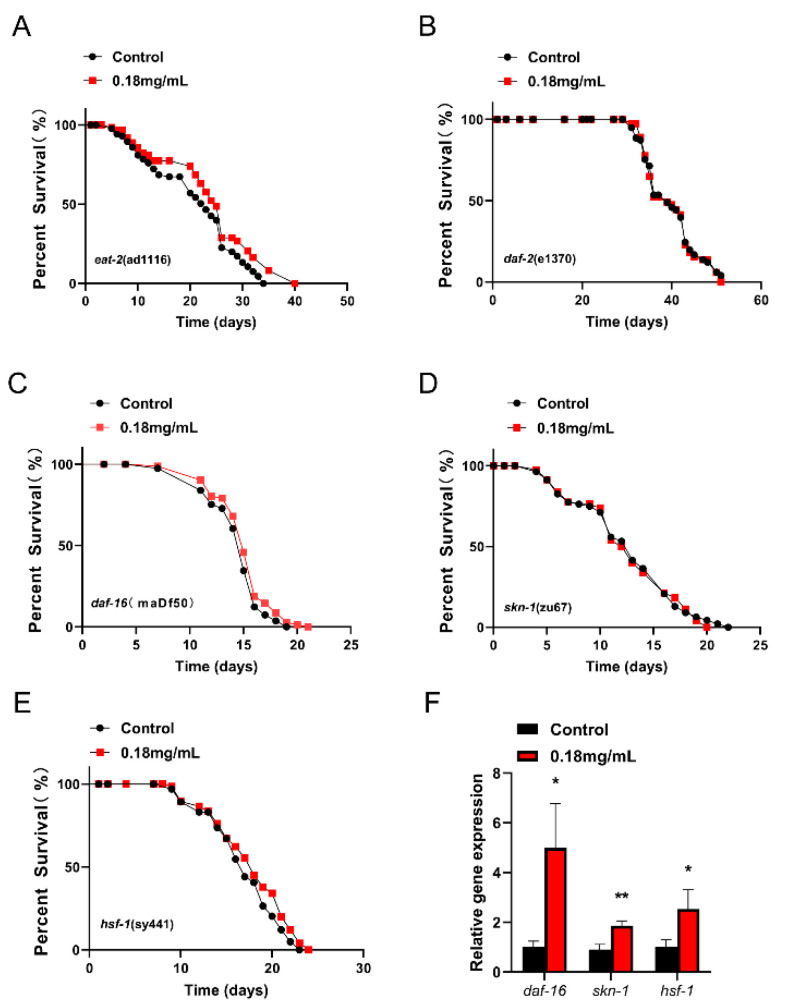
Longevity mechanism mediated by APPA. (**A**) APPA prolonged the lifespan of *eat-2* (ad1116). APPA treatment significantly increased the lifespan of *eat-2 (ad1116)*. *n* = 3 (100 individuals per group). (**B**) Effects of APPA treatment on lifespan of *daf-2 (e1370)*. *n* = 3 (100 individuals per group). (**C**) Effect of APPA treatment on lifespan of *daf-16 (mgDf50)*. *n* = 3 (100 individuals per group). (**D**) Effects of APPA treatment on the lifespan of *skn-1 (zu67)*. *n* = 3 (100 individuals per group). (**E**) Effects of APPA treatment on the lifespan of *hsf-1 (sy441)*. *n* = 3 (100 individuals per group). (**F**) *daf-16*, *skn-1* and *hsf-1* mRNA levels in nematodes treated with APPA. *n* = 3 (2000 individuals per group). Statistical analysis of the lifespan was performed using GraphPad 7 and *p* values were calculated by the log-rank test. Numerical data were analyzed by Student’s *t*-test and values were presented as mean ± SEM, * *p* < 0.05, ** *p* < 0.01.

**Figure 6 ijms-24-13682-f006:**
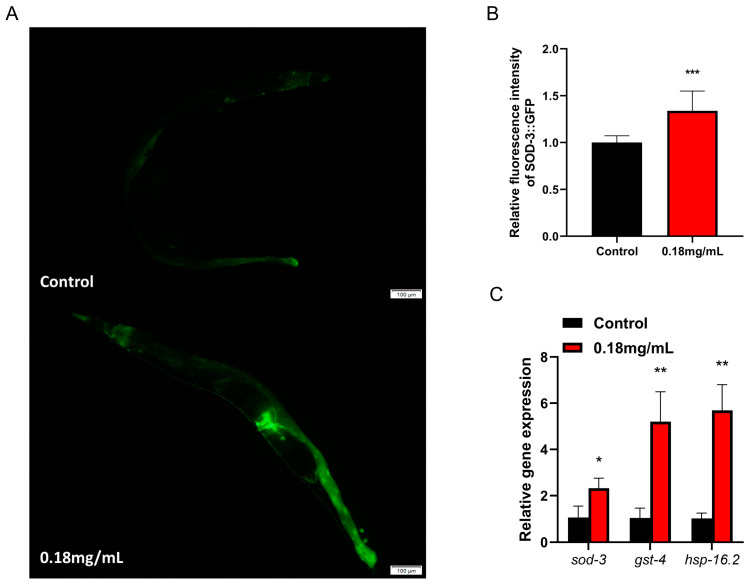
The APPA-mediated effect is associated with antioxidant genes. (**A**) Fluorescent image of CF1553. *n* = 3 (30 individuals per group). (**B**) Effect of APPA on the expression of SOD-3::GFP protein in nematodes (CF1553). APPA significantly induced expression of SOD-3::GFP. (**C**) The mRNA relative levels of *sod-3*, *gst-4*, and *hsp-16.2*, compared with control group. APPA significantly induced the expression of *sod-3*, *gst-4* and *hsp-16.2* mRNA. The images were analyzed with ImageJ software and numerical data were analyzed by Student’s *t*-test using GraphPad 7. *n* = 3 (2000 individuals per group) Values were presented as mean ± SEM, *** *p* < 0.001, ** *p* < 0.01, * *p* < 0.05.

**Figure 7 ijms-24-13682-f007:**
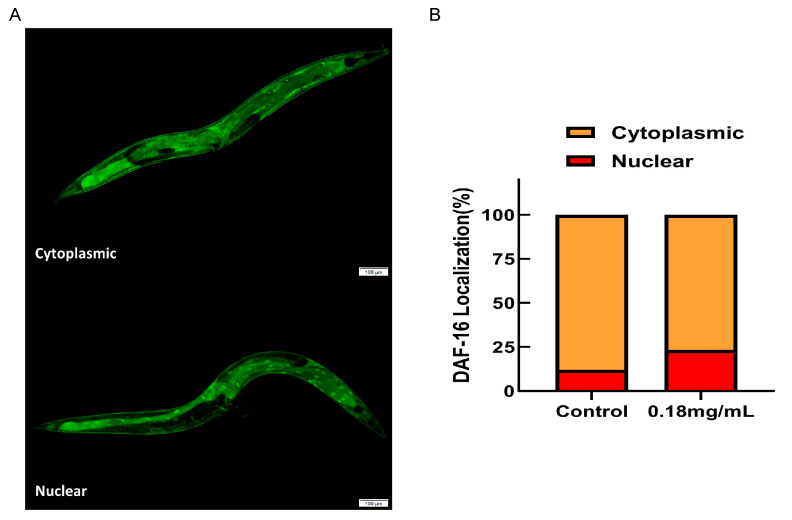
Effect of APPA on *daf-16* entry into nucleus. (**A**) Fluorescence microscopy was used to observe the nuclear translocation of DAF-16::GFP, indicating nuclear translocation of *daf-16* and cytoplasmic indicating no *daf-16* nucleation. *n* = 3 (30 individuals per group). (**B**) The proportion of the total number of nematodes admitted to the nucleus. Compared with the control group, APPA significantly increased the percentage of *daf-16* into the nucleus.

**Figure 8 ijms-24-13682-f008:**
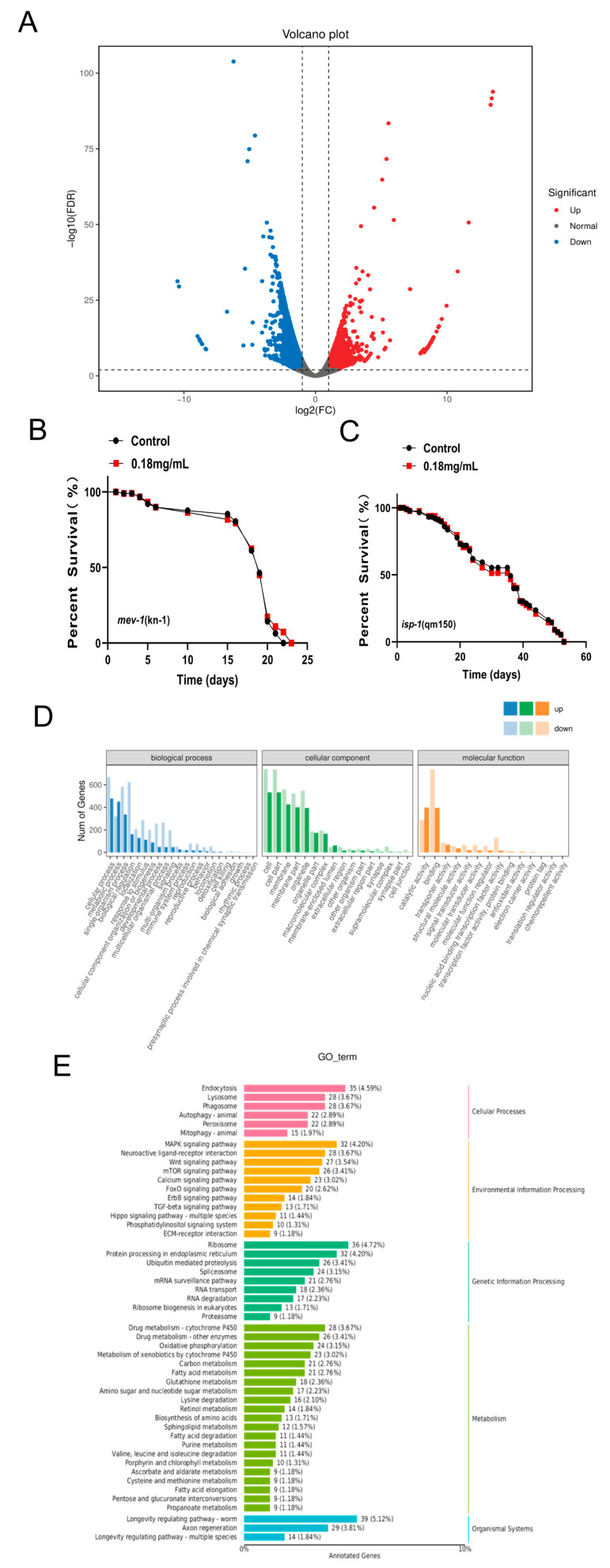
The effect of APPA on the regulation of nematodes and the mechanism by which APPA extends the lifespan of nematodes depend on the mitochondrial pathway. (**A**) There were 1548 upregulated genes and 2173 downregulated genes in the APPA-treated group (FDR < 0.01 and Fold Change ≥ 2). *n* = 3 (2000 individuals per group). (**B**) Effects of APPA on the lifespan of *mev-1 (kn1)*. *n* = 3 (30 individuals per group). (**C**) Effects of APPA on the lifespan of *isp-1 (qm150)*. *n* = 3 (30 individuals per group). (**D**) GO analysis of differential genes regulated by APPA treatment group. (**E**) KEGG analysis of differential genes in APPA-treated nematodes.

**Figure 9 ijms-24-13682-f009:**
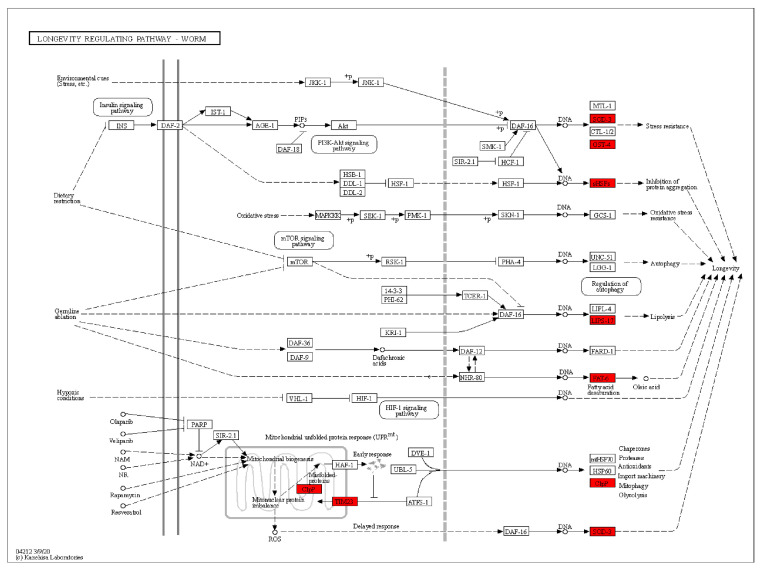
The KEGG analysis of differentially regulated genes.

## Data Availability

Not applicable.

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
