# Peer review of "APPA Increases Lifespan and Stress Resistance via Lipid Metabolism and Insulin/IGF-1 Signal Pathway in Caenorhabditis elegans"

_ijms, 2023, doi:10.3390/ijms241813682_

Round 1

Reviewer 1 Report

The article “APPA Increases Lifespan and Stress Resistance via Lipid Metabolism and Insulin/IGF-1 Signal Pathway in Caenorhabditis elegans” submitted to “International Journal of Molecular Sciences” provides interesting insights into the health- and lifespan promoting action of APPA and its molecular background mechanisms. The uncovering and understanding of compounds with anti-ageing and pro-longevity capacities is of general interest and well-selected methods were used to determine the effects on lifespan and healthspan in this study. Furthermore, several interesting genes were targeted via mutant strain analyses as well as in a qPCR assay. Finally, the RNAseq study enables deeper insights into the mechanisms of action of APPA. However, several issues should be addressed prior publication:

Abstract: Please shortly mention what APPA is and what the abbreviation stands for

Introduction:

·        Line 28: What does “Aging is worsening worldwide” mean? In which way? Please, write more precisely

·        Line 33: You wrote “The reactive oxygen species (ROS) generated in the body alters the cell shape and vitality, impairing metabolism and immunological function. Thus, telomere, mitochondrial, and DNA damage are directly related to aging”. I think the connection of these two sentences with “thus” is not correct. Please rewrite or add the information which are necessary to understand the “thus” in the sentence.

·        Line 46-55: Give a more detailed description about APPA (how is it synthesized, for what purpose is it used, are there any interesting previous results with APPA in other studies)

Results:

·        For all images and graphs: Please consider to add the number of used nematodes per assay to the figure legends.

·        Line 58-64: Please mention that further details for all survival assays are presented in supplementary table A3.

·        Line 71-75: You designated the measured autofluorescence as “Lipofuscin”, which is, however, not correct according to Pincus et al. 2016 (Autofluorescence as a measure of senescence in C. elegans: look to red, not blue or green. Aging (Albany NY), 8, 889). Please, replace “Lipofuscin” by “autofluorescence” throughout your manuscript or provide and discuss a recent study, which shows that Pincus et al. (2016) are wrong. Furthermore, according to Pincus et al. (2016), blue fluorescence is not a suitable setting to measure age-related autofluorescence in C. elegans. Thus, you should discuss the limited use of your results according to that paper.

·        Line 83-86: You wrote “According to the oil red O staining assay, fat accumulation in C. elegans tissues was drastically decreased under APPA treatment compared to the control, consistent with the findings beforehand, where APPA did not influence C. elegans growth and development but improved its motor function (Figure3A,B)”. The second part of the sentence is confusing. Why is that consistent? This needs to be explained in much more detail and, preferably, in the discussion part. And Figure 3 doesn’t show motor function, growth and development assays. Thus, it is unclear which results you discuss here.

·        Line 100-103: Endogenous oxidative stress was determined by the H2DCF-DA assay. Despite its frequent use, the H2DCF-DA assay was found to be unsuitable to measure intracellular ROS in C. elegans, which was affirmed by the editorial board of the journal “Free Radical Biology and Medicine” and by Labuschagne and Brenkman (2013) (Labuschagne CF, Brenkman AB (2013) Current methods in quantifying ROS and oxidative damage in Caenorhabditis elegans and other model organism of aging. Ageing Res Rev 12:918–930.) as well as Dikalov and Harrison (2014) (Dikalov SI, Harrison DG (2014) Methods for detection of mitochondrial and cellular reactive oxygen species. Antioxid Redox Signal 20:372–382.). Thus, the interpretation of the results should be done very carefully. Furthermore, the problems by using this assay should be discussed.

·        Line 114-119 / Figure 5: Is the paraquat assay not significant? No significance stars are given in the graph or are explained in the text.  

·        Line 128-131: You wrote “APPA treatment significantly increased C. elegans lifespan compared to the control group (Figure 6A, Table A3), consistent with the APPA reducing C. elegans swallowing frequency (Figure A1E)”. This is very confusing. Lifespan assay was already explained in chapter 2.1. What is the difference here? This needs to be explained in the text. And why do you assume that “that the mechanism of APPA's effect on C. elegans lifespan was unrelated to the dietary restriction.”. This also needs to be explained in much more detail.

Discussion:

·        In general: I miss the guiding thread in the discussion and the presentation of highlights. Maybe you could make sub-chapters? Furthermore, I miss a comparison with other (similar?) molecules with maybe similar or opposing action in C. elegans (Is APPA special or are there many substances with similar effects?). Finally, an outlook (How can the community use your results? What should be done next?) and the description of the limitations of your study would be great.

·        Line 132: Instead of “However, it did not affect” write “However, APPA did not affect”

·        Line 238: What does “Herein, APPA administration was done on the fifth day since that is when the improvement impact was the strongest.” mean? Does it mean that you always started the APPA treatment on the (about) second day of adulthood?

Methods:

·   For all imaging analyses: Please indicate which wavelength (or fluorescence filter set) and which microscope-magnification was used. Furthermore, please indicate how the measured intensity was normalized to the whole-body size of the worms (did you determine the whole-body size by using bright field images?). If you did not normalize the intensities, please consider re-analysing the photographs.

·   You did not perform the experiments in a blinded manner. This limits the objectivity of the presented results and this problem should be discussed in the article. In manually performed C. elegans analyses, especially lifespan or stress survival assays, researchers need to decide whether a worm is dead or alive by visual judgement. A certain expectation, such as an expected life prolonging ability of a substance, can have a substantial influence on this decision. Gruber et al. (2009) (Deceptively simple but simply deceptive–Caenorhabditis elegans lifespan studies: considerations for aging and antioxidant effects. FEBS Lett 583:3377–3387.) explained the operator bias in detail and suggest blinding and randomization especially for all survival studies in C. elegans.

·   Line 339-340: You wrote “The plates were incubated at 20°C for incubation, and every 24 hours.”. I guess, several words are missing here.

·   Line 359: Is there a reason to write “AGAR” with big letters?

·   Line 406-407: You wrote “Thirty mutant TJ356 nematodes were sedated with 10 mM leimidazole (Aladdin, Shanghai, China) and placed on a 2% agar-agar mat. Next, the nematodes were treated with 0.18 or 0 mg/mL of APPA for five days.” I guess, the sentences have the wrong order.

·        Line 412-422: Specify the PCR-cycles and temperatures used. Please indicate whether the PCR products were checked via gel electrophoresis and melting curve analysis.

·        Line 424: You wrote “The gene expressions in wild-type nematodes treated with 0.18 mg/mL of APPA at day five were assessed […]”. Does that mean, that you did not use an untreated control for the RNAseq assay but only the treated group? Please write more clearly. Furthermore, you should add some general information of the experimental setup (how many worms were used, how many biological replicates, how was the RNA isolated, mRNA enrichment step, size of cDNA fragments, did you select paired end sequencing, what was the read length and read depth)

Supplement:

Table A2: Add the used annealing temperatures, product sizes and the determined primer pair efficiencies for each primer pair. If you didn’t determine primer efficiencies, this limitation needs to be mentioned/discussed. Furthermore, only one reference gene was used in this study which limits the interpretation of the results. This should be also at least discussed or re-analysed using a second reference gene.

References:

Several references did not include the author’s names (such as reference number 40, 51, 54 etc.). Please check all references.

Reviewer 2 Report

Review on the manuscript of Wang, S. et al.: “APPA Increases Lifespan and Stress Resistance via Lipid Metabolism and Insulin/IGF-1 Signal Pathway in Caenorhabditis elegans”.

In this manuscript, authors explored the mechanisms involved in the extension of C. elegans lifespan by APPA.

The manuscript is clear and precise on the questions that authors proposed to answer. Thus, the issues that arise to me are listed below, so, I hope the authors find the following comments and suggestions useful.

1 – Authors observed a significant increase of C. elegans lifespan (3.49, 4.92, and 2.75%) following APPA treatment (0.09, 0.18, and 0.36 mg/mL of APPA, respectively). Is this increase in C. elegans lifespan biologically relevant?

2 – In Figure 3A, the control and APPA-treated animals have different sizes, which is indicative that animals were at different development stages (as authors claim that APPA treatment did not influence C. elegans growth and development). Can authors show representative images of animals at the same development stage?

3 - Can authors increase the information on the scale bar, as it is very difficult to read it in all figures with representative images?

4 – Authors observed that APPA elevated fat-6 and nhr-80 mRNA levels, as compared to control. Based on these observations, authors concluded that fat-6/nhr-80 significantly contributed to the APPA-mediated suppression of fat storage in C. elegans. Can authors clarify during the description of the data what fat-6 and nhr-80 genes are (this information only appears later in the discussion)? In addition, the upregulation of fat-6 and nhr-80 might be a consequence of reduced fat storage, but not the cause. Therefore, the conclusion taken by the authors may not be true.

5 – Can authors clarify the legend for Figure 3C (statistical analysis of the lifespan was performed using GraphPad 7 and p values were calculated by the log-rank test - is this information related to this figure?)

6 – Authors conclude that APPA increased the C. elegans longevity by reducing the ROS levels. Perhaps, the reason why C. elegans live longer following APPA treatment may not be related to the reduction in ROS levels.

7 – For data analysis, authors used two different versions (7 or 8) of the GraphPad Software. Is there any reason for that? I recommend authors to use the same GraphPad version for all statistical analysis.

8 – For the DAF-16 nuclear localization assay, can authors specify the type of cells that were used for analysis?

9 – Authors say that APPA was created and produced in the lab. Can authors provide the reference for the study that used APPA for the first time?

Reviewer 3 Report

There are many minor changes recommended, see below. Additional data must be added to figure 3. Statistics must be added/improved. Experiments describing mitochondrial content/function and catalase activity should be added. 

Line 10: APPA is not an acronym that I am familiar with. Maybe add “a benzida lysine analog” here too as you did in the introduction.

Line 15: the phrase “decreased the reactive oxygen species, lipofuscin, and fat accumulation,” should be changed to something like “decreased the levels of reactive oxygen species, lipofuscin, and fat,”

Line 16: I feel as if you shouldn’t start a sentence with a lowercase letter, even in the case of an abbreviation. I think this be changed to a capital Q, or rewritten so that you don’t begin with qRT-PCR.

Line 30: add a space before “Therefore …”

Line 30: add a space before “The reactive …”

Line 36: “Caenorhabditis elegans” should be italicized.

Line 40: “C. elegans” should be italicized.

Line 43: “C. elegans” should be italicized.

Line 44: “C. elegans” should be italicized.

Line 50: need one or more references for the sentence ending in “and animals.”

Line 52: probably should have a reference for the sentence ending in “in the lab.”

Line 53: need a reference for the sentence ending in “strong antioxidant action.”

Line 61: should remove the word “considerably …” A lifespan extension of 3-5% in worms does not seem considerable. Maybe use the word “maximally” here.

Line 65: need to include P-values somewhere in the figure caption or results section.

Line 77: Figure 2A is a very nice picture. However, it appears the fluorescence is decreased much more so (>50%?) than indicated in figure 2B (~10% decrease?). This should somehow be addressed in the results. Include the sample size in the figure too.

Lines 85 and 86: I’m seeing any data for growth or motor function in Figure 3A or 3B.

Line 86: same comment for qRT-PCR as for Line 16.

Line 86: be good to briefly define the function of fat-6 and nhr-80. Why are they good markers for fat synthesis or metabolism.

Line 97: need to add sample sizes.

Line 109: need to add sample sizes.

Lines 111-121: these paragraphs seem a bit convoluted and could be simplified somewhat.

Lines 123-126: Figure 5 caption – need to include sample sizes. Also, I don’t see how the paraquat data was analyzed by Student’s t test. There are no error bars or asterisks. Maybe a Kaplan–Meier survival test was used?

Lines 127-131: there is no mention of eat-2 in the results text despite showing data in Figure 6A.

Lines 151-158: Figure 6 - need to include the tests used, sample sizes, and P-values for panels A-E.   

Line 159: Section 2.10 – seems like it would be good to also include catalase expression too as it is a major antioxidant enzyme.

Line 160: same comment for qRT-PCR as for Line 16.

Line 185: don’t like the phrase “Daf-16’s flaws” here. Maybe say “Daf-16 deficiency …”   

Line 191: to use the word “significantly” you need to show some statistics.

Line 211: Figure 9 is too small to read clearly – needs to be bigger.

Lines 219-220: since “mitochondria-related pathways were 219 dramatically altered” it seems that the paper could benefit from some mitochondrial assays, such as mitochondrial content and oxygen consumption.  

Line 237: this data needs to be included in the manuscript figures.

Line 239: shouldn’t start a sentence with an abbreviation.

Line 265: add a space in front of “APPA ...”

Line 269: add a space in front of “Here ...”

Line 271: shouldn’t start a sentence with an abbreviation.

Line 284: what is meant by “sub-death?”

Line 293: “C. elegans” should be italicized.

Line 295: add a space in front of “The increase ...”

English is mostly good, but there are some areas (indicated in the other comments) that need to be improved. 

Round 2

Reviewer 3 Report

I think you have addressed my comments/criticisms to a sufficient level.